

# Measurement of scattering and absorption properties of dust aerosol

# in a Gobi farmland region of northwest China—a potential

# anthropogenic influence

Jianrong Bi, Jianping Huang, Jinsen Shi, Zhiyuan Hu, Tian Zhou, Guolong Zhang,

Zhongwei Huang, Xin Wang, and Hongchun Jin

Key Laboratory for Semi-Arid Climate Change of the Ministry of Education, College of

Atmospheric Sciences, Lanzhou University, Lanzhou 730000, China

*Correspondence to:* Jianping Huang (hjp@lzu.edu.cn)

**Abstract.** We conducted a comprehensive field campaign on exploring the optical

characteristics of mineral dust in Dunhuang farmland nearby the Gobi deserts of

northwest China during spring of 2012. The day-to-day and diurnal variations of dust

aerosol showed prominent features throughout the experiment, primarily attributable

to frequent dust events and local anthropogenic emissions. The overall average mass

concentration of the particulate matter with an aerodynamic diameter less than 10 μm

($PM_{10}$), light scattering coefficient ($\sigma_{sp,670}$), absorption coefficient ($\sigma_{ap,670}$), and

single-scattering albedo ($SSA_{670}$) were 113±169 μgm$^{-3}$, 53.3±74.8 Mm$^{-1}$, 3.2±2.4

Mm$^{-1}$, and 0.913±0.05, which were comparable to the background levels in southern

United States, but smaller than that in the eastern and other northwestern China. The

anthropogenic dust produced by agricultural cultivations (e.g., land planning, plowing,

and disking) exerted a significant superimposed effect on high dust concentrations in

Dunhuang farmland prior to the growing season (i.e., from 1 April to 10 May). Strong

south valley wind and vertical mixing in daytime scavenged the pollution and weak

northeast mountain wind and stable inversion layer at night favorably accumulated the

air pollutants near the surface. In the afternoon (13:00–18:00 LT), mean $SSA_{670}$ was

0.945±0.04 that was predominant by dust particles, whereas finer particles and lower





$SSA_{670}$ values (~0.90–0.92) were measured at night, suggesting the potential
influence by the mixed dust-pollutants. During a typical biomass burning event on 4
April 2012, $\sigma_{ap,670}$ changed from ~2.0 $Mm^{-1}$ to 4.75 $Mm^{-1}$ and $SSA_{670}$ changed from
~0.90 to ~0.83, implying remarkable modification of aerosol absorptive properties
induced by human activities. The findings of this study would help to advance an
in-depth understanding of the interaction among dust aerosol, atmospheric chemistry,
and climate change in desert source region.

## 1. Introduction

Asian mineral dust (also known as dust aerosol) in the atmosphere is deemed to
exert a profound impact on air quality and climate change. It can perturb the energy
budget of the Earth system directly through scattering and absorption of solar and
terrestrial radiation (Huang et al., 2009, 2014; Ge et al., 2010; Li et al., 2016) and
indirectly by altering cloud microphysical processes and related hydrological cycle
(Rosenfeld et al., 2001; J. Huang et al., 2005, 2006, 2010; Yin and Chen, 2007; W.
Wang et al., 2010; Creamean et al., 2013; Wu et al., 2016), as well as modifying snow
and ice surface albedo (Aoki et al., 2006; Huang et al., 2011; Wang et al., 2013; Qian
et al., 2014). In addition, alkaline mineral dust carries abundant organic matters and
iron ions deposited on the surface of earth, and hence affects biomass productivity of
the North Pacific Ocean and relevant atmosphere-ocean carbon exchange, which
plays a pivotal role in the global biogeochemical cycle and carbon cycle (Cao et al.,
2005; Jickells et al., 2005; Maher et al., 2010; Shao et al., 2011).
The Taklimakan Desert in northwestern China and Gobi Deserts in southern
Mongolia and northern China are widely regarded as two major active centers of dust
storms in East Asia (Sun et al., 2001; Zhao et al., 2006; Wang et al., 2008; Ge et al.,
2016). These extensive arid and desert zones frequently generate a great deal of tiny
soil particles every spring that are uplifted and entrained into the free atmosphere
layer via cold frontal cyclones (Zhang et al., 1997; Aoki et al., 2005; Kai et al., 2008;
J. Huang et al., 2009, 2010, 2014). Affected by mid-latitude prevailing westerlies,
these dust particles can transport long distances on a subcontinental scale, even sweep





across the remote Pacific Ocean and occasionally arrive at the west coast of North
America during the peak seasons of strong dust storms (Zhao et al., 2006; Uno et al.,
2009, 2011). They then have a far-reaching influence on climatic and environmental
changes both regionally and globally. Until now, there have been a large number of
intensive field experiments (e.g., ACE–Asia, ADEC, PACDEX, EAST–AIRC) and
ground-based aerosol monitoring networks (e.g., AERONET, SKYNET, CARSNET)
for probing the Asian mineral dust (Holben et al., 1998; Huebert et al., 2003;
Nakajima et al., 2003; Takamura et al., 2004; Eck et al., 2005; Mikami et al., 2006;
Huang et al., 2008a; Che et al., 2009, 2015; Li et al., 2011), which is crucial to aid in
thoroughly understanding the climatic effects of dust aerosols over East Asian domain.
Nevertheless, due to poorly sampled over desert source areas of northwest China, the
light scattering and absorption properties of mineral dusts in this region are far
inadequate and urgently need to be further surveyed.
The Intergovernmental Panel on Climate Change (IPCC, 2013) reported that the
symbol and magnitude of the radiative forcing of mineral dust is greatly reliant on the
accurate and reliable knowledge of aerosol total loading, microphysical and chemical
characteristics, as well as its spatiotemporal distribution. The current consensus is that
nearly pure dust aerosol in the globe has relatively low light-absorption, with
single-scattering albedo of ~0.96–0.99 (Dubovik et al., 2002; Anderson et al., 2003;
Uchiyama et al., 2005; Bi et al., 2014, 2016), which principally depends on the
fraction and mixing ways of ferric iron oxides (i.e., hematite and goethite) in dust
(Sokolik and Toon, 1999; Lafon et al., 2004, 2006). However, the coexistences of
both mineral dust and other types of aerosols originated from diverse human activities
(e.g., coal combustion, mobile source emissions, and biomass burning) are ubiquitous
in the real atmosphere, which increases the complexity and variability to aerosol key
parameters (Arimoto et al., 2004; Xu et al., 2004; Wang et al., 2015). When the lofted
dust plumes in desert source areas are traveled eastward across the polluted regions,
they commonly mix with anthropogenic pollutants and enhance heterogeneous
chemical reactions with other reactive gas species, and then may remarkably alter
their chemical and microphysical properties (Arimoto et al., 2006; Li and Shao et al.,





2009; Nie et al., 2014). It is well documented that the mineral dust might have already
mixed with polluted aerosols in near dust source regions of northwest China (i.e.,
Mongolia Gobi desert), besides the mixing processes on the transport pathway (K.
Huang et al., 2010). Xu et al. (2004) indicated that both dust aerosol and local
pollution sources coexisted in Yulin nearby the Mu Us desert of northwest China
during April 2001, which produced a significant influence on aerosol properties in the
region. Likewise, Li et al. (2010) analyzed trace gases and aerosols observed at
Zhangye (39.082°N, 100.276° E, 1460 m above MSL), a rural site within the Hexi
Corridor in northwest China during spring 2008, and uncovered that the mixing
between mineral dust and anthropogenic air pollutants can be omnipresent in this area,
including at nighttime or during severe dust events. It implies that prior to moving out
from the source region, dust particles were likely in connection with pollutants. For
the sparsely populated and lesser anthropogenic affected desert source regions in
northwest China (e.g., the Taklimakan Desert and its adjacent areas), the interaction
between local pollutions and mineral dust is deserved to explore in depth. This is of
prime importance to ascertain the relative contributions of two different aerosol
sources in atmospheric chemistry and regional climate change.

To advance a better understanding of the drought processes and dust-relevant

climatic impacts in northwest China (Huang et al., 2008b; Bi et al., 2011; G. Wang et
al., 2010), the Semi-Arid Climate and Environment Observatory of Lanzhou
University (henceforth referred to as SACOL, http://climate.lzu.edu.cn/english/)
carried out a comprehensive field campaign in Dunhuang during spring of 2012.
Dunhuang is situated at the westernmost fringe of Hexi Corridor in Gansu province,
close to the east edge of Kumtag Desert and about 450 km in the downwind zone of
Taklimakan Desert. It is an important town of the ancient Silk Road and the
transportation junction to the ancient western region, central Asia and Europe, which
has become a world-famous tourist city with a residential population of 200,000. The
agriculture and tourism are the dominant economic industries in Dunhuang. An array
of ground-based remote sensing and in situ instruments were set up during the
intensive period, which sought to investigate the aerosol key properties and its





climatic effect on regional scale (Bi et al., 2014). This study especially aims at
exploring the light scattering and absorption characteristics of mineral dust and
elucidates a potential anthropogenic influence. In the following, we first introduce the
site information and integrated measurements in Section 2. The primary results and
discussion are described in Sect. 3. The concluding remarks are given in Sect. 4 and
followed by the data availability in Sect. 5.

## 2. Site and instrumentation

### 2.1. Site information

SACOL's Mobile Facility (SMF) was deployed at Dunhuang farmland (40.492° N,
94.955° E, 1061 m above MSL) from 1 April to 12 June 2012. The site is a tiny
isolated oasis encompassed by east-west oriented Gobi desert and arid zones in
northwest China, with the Mingsha Shan (Echoing-Sand Mountain, elevation: ~1650
m) and Sanwei Mountain (elevation: ~1360 m) to the southeast, and the Beishan
Mountain (elevation: ~2580 m) to the north (Ma et al., 2013). The underlying surface
is typically covered with Gobi desert and saline-alkali land, and the principal
vegetation types consist of extremely sparse Alhagi. Dunhuang farmland is an
important agricultural base in Gobi desert, mainly planting hami melon and cotton.
There are not any significant manmade pollution sources (e.g., large-scale industries
or coal-fired power plants) around the monitoring station. The southwest-northeast
oriented National Highway 215 is about 400 m away from the west of the site (Figure
1a). The nearest Xihu township (with total population of 13,800) is approximately 7
km to the north of Dunhuang farmland, along with some scattered villages stretching
from west to east. Meanwhile, the station is located in the northeast of Dunhuang city
(~45 km), at the west of Guazhou country (~70 km), and at the southwest of Liuyuan
town (~80 km). In general, the major anthropogenic emission sources at Dunhuang
farmland likely include coal combustion from domestic heating and cooking, mobile
sources emissions from vehicle exhaust gas, and biomass burning from crop residue
and traditional ritual activities, which are ordinarily considered to be a puny
contribution to the mineral dust in present-day climate models. The climate pattern



here is characterized as extreme drought but with a moderate temperature during the
whole sampling period (temperature: 18.3±8.1℃, relative humidity, RH: 21.9±16.5%,
mean ± standard deviation). Thereby the dust storms frequently take place in this
region from spring to early summer. Figure 1(b) shows the overall mean UV aerosol
index (AI) from 1 April to 12 June 2012 obtained from the Ozone Monitoring
Instrument (OMI) absorbing aerosol products (Torres et al., 2007). The AI dataset is a
very good indicator for mapping the distribution of absorbing aerosols (mainly black
carbon and dust). High AI values (>0.7) distributions are well consistent with the
dust-dominated geomorphological features in arid and semiarid regions (i.e.,
Taklimakan Desert and Gobi deserts). It is very obvious that Dunhuang (marked with
a pentagram) is also situated at the primary dust belt of northwest China, as presented
in Figure 1b.

**2.2. Aerosol measurements**

An aerosol integrated observing system is installed in the laboratory of SMF and
utilized to continuously measure aerosol optical properties and size distribution in the
field. Prior to the experiment, the in-situ aerosol instruments and broadband
radiometers were newly purchased and calibrated by the manufacturers (Bi et al.,
2014). Table 1 summarizes the basic specification, measured variables, and accuracy
of surface-based instruments deployed at Dunhuang farmland throughout the
experiment. We shall describe sequentially as below.
An ambient particulate monitor (Model RP1400a, Rupprecht and Patashnick
Corp.) can collect the in situ mass concentration of the particulate matter with an
aerodynamic diameter less than 10 μm ($PM_{10}$) based on Tapered Element Oscillating
Microbalance (TEOM) technique. The measurement range and accuracy of $PM_{10}$
concentration levels are normally 0–5 $gm^{-3}$ and 0.1 $\mu gm^{-3}$, respectively. The heating
temperature (~50℃) of the sampling tube may cause a partial loss of volatile and
semivolatile aerosol compounds and hence bring about a negative signal. In this study,
we eliminate all the negative values of $PM_{10}$ concentrations, which account for less
than 1% of total data points.
An integrating nephelometer (Model 3563, TSI Inc.) is designed to simultaneously



measure the total scattering coefficients ($\sigma_{sp}$) and hemispheric backscattering
coefficients ($\sigma_{bsp}$) of aerosol particles at three wavelengths of 450, 550, and 700 nm,
with the $\sigma_{sp}$ detection limits of 0.44, 0.17, and 0.26 $Mm^{-1}$ (1 $Mm^{-1}=10^{-6}$ $m^{-1}$),
respectively (signal-to-noise ratio of 2) (Anderson et al., 1996). To quantify the
instrument drift and improve accuracy, we periodically perform the routine calibration
using air and high-purity $CO_2$ gases. Furthermore, the truncation errors of
near-forward scattering (i.e., nonideal angular effects) are corrected according to the
method of Anderson and Ogren (1998). The observed ambient RH values are mostly
smaller than 40% throughout the entire period. It is well-documented that RH-induced
the variations in aerosol light scattering coefficients are minimized under a low
sampling stream RH of 10–40% (Covert et al., 1972). In this paper, we computed the
scattering Ångström exponent at 450–700 nm (SAE 450/700 nm) from $\sigma_{sp}$ at 450 nm
and $\sigma_{sp}$ at 700 nm by utilizing a log-linear fitting algorithm. And thus $\sigma_{sp}$ at 670 nm
($\sigma_{sp,670}$) was logarithmic interpolated between $\sigma_{sp,450}$ and $\sigma_{sp,700}$.
A multi-angle absorption photometer (MAAP Model 5012, Thermo Electron
Corp.) is capable of observing the aerosol light absorption coefficient at 670 nm
($\sigma_{ap,670}$) by filter based methods without requirement of post-measurement data
correction or parallel-measured aerosol light-scattering coefficients (Petzold et al.,
2002). The instrument detects an emitted light at 670 nm in the forward and back
hemisphere of airborne aerosols deposited on a fiber filter, which is used to improve
multiple scattering effects in the aerosol optical properties via a radiative transfer
scheme (Petzold et al., 2002, 2005). The sample flow rate is 1000 L/h, with flow error
of < 1%. We made use of a specific absorption efficiency at 670 nm of 6.5±0.5 $m^2g^{-1}$
to estimate black carbon (BC) concentration from $\sigma_{ap,670}$ as recommended by Petzold
et al. (2002).
An Aerodynamic Particle Sizer (APS) spectrometer (Model 3321, TSI Inc.) can
continuously provide the real-time, high-resolution aerosol size distribution with
aerodynamic diameters from 0.5 to 20 μm range (52 channels). When extreme dust
episodes outbreak, an aerosol diluter (Model 3302A, TSI Inc.) is operated in series
with APS to reduce particle concentrations in high-concentration aerosols, which





offers a representative sampling that meets the input requirements of the APS
spectrometer. All the mentioned-above aerosol datasets were acquired at 5-minute and
hourly averages, and reported for sampling volumes under standard air conditions (i.e.,
1013.25 hPa and 20 ℃).
**2.3.  Other ground-based measurements**
A Micro-Pulse Lidar (Model MPL–4, Sigma Space Corp., U.S.A.) is a compact
and unattended apparatus for providing continuous data information of extinction
coefficient and depolarization ratio profiles of aerosols and clouds (Welton et al.,
2000). The MPL–4 emits a laser beam at 527 nm wavelength from a Nd:YLF pulsed
laser diode and receives the attenuated backscattering intensity and depolarized
signals from aerosol particles or cloud droplets with a 30-meter vertical resolution and
a 1-minute average interval. And we can acquire the accurate backscattering profile
by means of a series of corrections (e.g., dead time, background signal, afterpulse,
overlap, and range-corrected) according to the standard methods (Campbell et al.,
2002). The detailed data acquisition and retrieval algorithms of the lidar system can
be referred to the publications of Campbell et al. (2002) and Z. Huang et al. (2010).
A weather transmitter (Model WXT–520, Vaisala, Finland) is set up on the top of
the SMF trailer and recorded the air temperature (T in ℃), relative humidity (RH),
ambient pressure (P, unit: hPa), wind speed and wind direction at 20 seconds interval.
In this article, we calculated the 5-minute and hourly averages from the raw data.
A dozen of state-of-the-art broadband radiometers were installed in a row on a
standard horizontal platform (~4 m above the surface) where the field of view was
unobstructed in all directions (Bi et al., 2014). The direct normal irradiance and
diffuse irradiance were independently measured by an incident pyrheliometer (Model
NIP, Eppley Lab.) and by a ventilated and shaded pyranometer (Model PSP, Eppley
Lab.), which were mounted on a two-axis automatic sun tracker (Model 2AP,
Kipp&Zonen). The global irradiance (0.285–2.8 μm) and downward longwave
irradiance (3.5–50 μm) were respectively gathered from a ventilated PSP pyranometer
and a ventilated and shaded pyrgeometer (Model PIR, Eppley Lab.). All irradiance
quantities were stored in a Campbell data logger with 1-minute resolution.



Additionally, a Total Sky Imager (Model TSI–880, YES Inc.) provides high-resolution
sky pictures every one minute during the daytime, which can detect and identify the
important weather conditions, such as dust storm, smoky pollution, rainy day, cloudy
or cloudless days.
**2.4. MERRA reanalysis products**
The MERRA (Modern–Era Retrospective Analysis for Research and Applications)
reanalysis assimilates a variety of conventional observations (i.e., temperature,
pressure, height, wind components) from surface weather stations, balloons, aircraft,
ships, buoys, and satellites from 1980 to the present, which is primarily committed to
improve upon the hydrologic cycle and energy budget for the science community
(Rienecker et al., 2011). In this paper, we took advantage of the 6-hourly average
wind fields at 500 hPa and 850 hPa levels from the MERRA reanalysis products.
**3. Results and discussion**
**3.1 Aerosol optical properties**
The aerosol single-scattering albedo (SSA) at 670 nm is defined as the ratio of the
light scattering coefficient ($\sigma_{sp,670}$) to the total extinction coefficient (the sum of $\sigma_{sp,670}$
and $\sigma_{ap,670}$). The SSA reflects the absorptive ability of aerosol particle and is a key
quantity in determining the sign (warming or cooling) of aerosol radiative forcing
(Hansen et al., 1997; Ramanathan et al., 2001).
Figure 2 delineates the time series of hourly average $PM_{10}$ mass concentration,
aerosol optical properties and size distribution at Dunhuang farmland during the
whole period. The overall mean, standard deviation, median, and different percentiles
of aerosol optical properties are also tabulated in Table 2. Aerosol optical features
exhibit dramatic day-to-day variations at Dunhuang. It is apparent that aerosol
loadings in April and early May are systematically higher than that in late May and
June, which agrees well with the results of columnar aerosol optical depths derived
from sky radiometer (Bi et al., 2014). This is chiefly attributed to the invading mineral
particulates from the frequent occurrences of intense dust storms in spring season.
The highly unstable synoptic cyclones (i.e., Mongolia cyclones) are regularly





hovering about the northern China and Mongolia in springtime, which trigger
high-frequency strong surface winds (Sun et al., 2001; Shao et al., 2011). The rising
temperature in this season leads to the melting of frozen soil and snow cover, leaving
behind a loose land surface and abundant bare soil sources, therefore affords a
favorable condition for dust storms. In addition, the contributions of local dust
emissions couldn't be ignored. We have clearly recorded that there were numerous
agricultural cultivated operations (e.g., land planning, plowing, and disking)
throughout the Dunhuang farmland district from 1 April to 10 May, which produced a
great amount of agricultural soil particles under strong winds, and thus had a
significant superimposed effect on elevated dust loading in the source and downwind
regions prior to the growing season. Those dust aerosols originated from disturbed
soils induced by human activities are interpreted as anthropogenic dust (Tegen and
Fung, 1995). Recently, some investigators estimated that anthropogenic dust could
account for approximately 25% of the global dust load (Ginoux et al., 2012; Huang et
al., 2015), and more than 53% of the anthropogenic sources mostly came from
semi-arid and semi-wet zones (Huang et al., 2015; Guan et al., 2016). Nonetheless, it
still remains a challenging task to distinguish between the natural and anthropogenic
fractions of mineral dust by employing a onefold technology, for instance, laboratory
analysis, in situ measurements, model simulations, active and passive remote sensing
methods (e.g., multichannel lidar, sun/sky radiometer), which should be combined
together (Bi et al., 2016). The overall mean $PM_{10}$ concentration was $113\pm169$ $\mu gm^{-3}$
(mean ± standard deviation), which is ~39% lower than the $184.1\pm212$ $\mu gm^{-3}$ average
level in Dunhuang (40.1° N, 94.6° E, 1139 m) during the spring of 2004 (Yan, 2007),
and ~26% smaller than the value of $153\pm230$ $\mu gm^{-3}$ measured at Zhangye (39.082° N,
100.276° E, 1460 m) during spring of 2008 (Li et al., 2010). Wang et al. (2015)
obtained a total average $PM_{10}$ concentration of $172\pm180$ $\mu gm^{-3}$ at SACOL during late
spring of 2007 (from 25 April to 25 June). And the mean $PM_{10}$ levels at Hunshan
Dake sandland in northern China during spring of 2001 varied between 226 and 522
$\mu gm^{-3}$ (Cheng et al., 2005).
The hourly average aerosol light scattering coefficient at 670 nm ($\sigma_{sp,670}$) was
$53.3\pm74.8$ Mm$^{-1}$. The big standard deviations of PM$_{10}$ and $\sigma_{sp}$ are possibly associated
with the injection of dust particles during the intense dust storms. Our result was
about a factor of 3 lower than the $\sigma_{sp}$ at 500 nm in mentioned-above other sites over
northern China (i.e., $126\pm90$ Mm$^{-1}$ for Dunhuang, $159\pm191$ Mm$^{-1}$ for Zhangye,
$164\pm89$ Mm$^{-1}$ for SACOL). Despite relatively small magnitude, the aerosol light
absorption coefficient at 670 nm ($\sigma_{ap,670}$) also presented pronounced variations, with
an average value and a maximum of $3.2\pm2.4$ Mm$^{-1}$ and 25.0 Mm$^{-1}$, respectively. This
result was a factor of 2 smaller than Yulin ($6\pm11$ Mm$^{-1}$) nearby Mu Us desert in
northwest China (Xu et al., 2004), and a factor of 5~7 far less than that at Shangdianzi
rural site ($17.5\pm13.4$ Mm$^{-1}$) in northern China (Yan et al., 2008) and Lin'an site (~23
Mm$^{-1}$) in southern China (Xu et al., 2002). The mean light scattering and absorption
coefficients in this study are comparable to the background levels (~$46.9\pm16.9$ and
$2.5\pm1.1$ Mm$^{-1}$) in Southern Great Plain of U.S.A (Delene and Ogren, 2002). This
suggests that extremely low levels of light absorption and scattering substances are
widely distributed throughout the Dunhuang region during the spring of 2012.
Therefore, a little perturbation stemmed from human activities (e.g., agricultural
cultivation, coal combustion from domestic heating and cooking, and biomass burning)
would undoubtedly exert a considerable impact on the light absorption property.

A few of strong dust episodes (4, 21–22, and 30 April, 1–3, 8–11, and 20 May, 4

and 10 June, corresponding to DOY 95, 112–113, 121, 122–125, 129–132, 141, 156,
and 162) could remarkably elevate the hourly average values of PM$_{10}$, $\sigma_{sp}$, $\sigma_{ap}$, and
aerosol size distribution (see Figure 2). During these dust events, the hourly PM$_{10}$
concentrations generally exceeded 1000 $\mu$gm$^{-3}$ and even approached 2000 $\mu$gm$^{-3}$,
which were tenfold greater than the overall mean level. The hourly $\sigma_{sp}$ were more
than 400 Mm$^{-1}$ or even close to 800 Mm$^{-1}$, and the corresponding $\sigma_{ap}$ varied between
10 Mm$^{-1}$ and 25 Mm$^{-1}$. Moreover, the peak values of aerosol number size distribution
concentrated in the particle diameters of 1–3 $\mu$m, which was consonant with the result
from remote sensing (Bi et al., 2014, 2016).

Figure 3 depicts the time evolutions of MPL normalized relative backscatter and

depolarization ratio at Dunhuang farmland from 1 April to 12 June 2012. The



depolarization ratio ($\delta$) is a useful indication to discriminate between spherical
particles ($\delta$ of ~0–0.1) and nonspherical particles (mainly dust aerosol, $\delta$ >0.2), since
it is very sensitive to the nonsphericity of scattering particle (Kobayashi et al., 1985;
Murayama et al., 1999; Shimizu et al., 2004; Huang et al., 2015). From Figure 3, we
can distinctly see that there was always a dense dust layer appeared at a height below
2 to 4 km during the whole experiment, with the peak value centered on 1.0–1.5 km,
which was within the planetary boundary layer (PBL). And the $\delta$ values commonly
reached above 0.3 (> ~0.3–0.5) during the heavy dust events and varied between 0
and 0.1 under clear-sky conditions (e.g., 6–7 April, 14–15 and 29 May, 9 June).
**3.2 Diurnal variations**
Figure 4 illustrates the diurnal variations of wind vector (ms$^{-1}$), air temperature (T
in ℃), RH (%), PM$_{10}$ ($\mu$gm$^{-3}$), $\sigma_{sp,670}$ (Mm$^{-1}$), $\sigma_{ap,670}$ (Mm$^{-1}$), aerosol number size
distribution (dN/dlogD in cm$^{-3}$), SAE at 450–700 nm, and SSA at 670 nm in
Dunhuang farmland from 1 April to 12 June 2012. Note that the APS spectrometer
was operated from 30 May to 12 June. A discernible wind vector was showed in the
diurnal variation, in other words, strong southwest wind and south wind dominated in
the daytime, from 11:00 to 24:00 LT (local time), and transformed into the weak
northeast wind prevailed from the midnight to the following morning of 10:00 LT.
The prominent phenomenon can be roughly interpreted by classical mountain-valley
wind circulation, which was primarily generated by the diurnal differences of
temperature between the mountain slope and the valley floor. During the daytime, the
huge Beishan Mountain slope heats up by the solar radiation more rapidly than the
valley floor, which causes convection above the mountain slope. The compensating
airflow is consequently directed toward the mountain slope, inducing upslope
southerly wind, or the valley wind, which usually peaks near midday and gradually
disappears after sunset. Conversely, at night, radiative cooling of the mountain slope
is more quickly than the valley floor, inducing the mountain wind, which generally
reaches maximum strength just before sunrise (Arya, 1999). Throughout the
experiment, air temperature displayed a large diurnal variation (with the diurnal
difference of $\delta$T~26 ℃) and RH always kept below 40% for the whole day. It is very





clear that the minimal T and maximal RH arose at around 06:00–07:00 LT, and the
maximal T and minimal RH occurred at about 16:00 LT, which represented an
energetic vertical turbulent motion in daytime and a stable radiative temperature
inversion during nighttime.
The aerosol optical parameters also exhibited striking diurnal variations, which
were closely related to the local meteorological elements. During the daytime
(10:00–18:00 LT), the $PM_{10}$ concentration remained high level (~57–65 $\mu gm^{-3}$) and
increased sharply from 19:00 LT and reached a maximum of 84.2 $\mu gm^{-3}$ at 20:00 LT.
The $PM_{10}$ began to decrease from 21:00 LT to the next morning. A low level (~40–46
$\mu gm^{-3}$) kept in the midnight (00:00–05:00 LT) and rose gradually from 06:00 LT and
attained a secondary peak value of 55.7 $\mu gm^{-3}$ at 07:00 LT. The aerosol light
scattering ($\sigma_{sp,670}$) presented a similar pattern with $PM_{10}$, but the maximal value (~42
$Mm^{-1}$) appeared at 13:00 LT, with the other two secondary peak values occurred at
20:00 (~34.1 $Mm^{-1}$) and 07:00 LT (~27.3 $Mm^{-1}$). The high levels of $PM_{10}$ and $\sigma_{sp}$
during the daytime were primarily attributable to strong south wind from Gobi region
and local dust emissions. By contrast, aerosol light absorption coefficient ($\sigma_{ap,670}$)
showed a more pronounced diurnal feature, which was well proved to be majorly
controlled by anthropogenic emissions (Li et al., 2010). The diurnal $\sigma_{ap}$ always stayed
at a low level (~2.0 $Mm^{-1}$) from 13:00–18:00 LT, and also reached a maximum of 3.3
$Mm^{-1}$ at 20:00 LT. Subsequently, $\sigma_{ap}$ dramatically reduced from midnight and
preserved at a low value of about 2.2 $Mm^{-1}$ from 02:00–04:00 LT, and remained a
steadily high level of ~2.7–2.9 $Mm^{-1}$ from 05:00–10:00 LT. It was probably explained
as follows. The influences of local anthropogenic pollutants were commonly small in
the afternoon, because the strong southerly wind from Gobi deserts and powerful
daytime vertical convection mixing efficiently dilute local air pollutants. Whereas
weak northeast wind and stable temperature inversion at night facilitate the
accumulation of pollutants within the PBL, hence nighttime levels were normally
larger. Increasing human activities (e.g., domestic cooking, traffic emissions for
transportation and agriculture) in the early morning might also be responsible for the
morning peaks in the aerosol absorption coefficient. The $\sigma_{ap}$ maximum at 20:00 LT





was presumably influenced by the mixture of mineral dust and anthropogenic
pollutants. This conclusion could be partly supported by the diurnal variation of SAE
at 450–700 nm (Figure 4), which showed high SAE values (~0.5–0.6) appeared at
02:00–10:00 LT and low SAE (~0.2–0.3) occurred on 13:00–22:00 LT. Generally,
large SAE around 0.6 represents small particles (e.g., urban-polluted aerosol or soot)
and low SAE less than 0.3 or negative value corresponds to coarse-dominated large
size particles (e.g., dust or sea salt) (Anderson et al., 2003).
Furthermore, aerosol number size distribution exhibited a noticeable supermicron
particles dominated in the entire day, probably linked to the predominant dust aerosol
in daytime and local anthropogenic emissions at nighttime. In this study, we
postulated that the aerosol light extinction at shortwave waveband is completely
caused by those particles with aerodynamic diameters of 10 μm or less. And the mass
scattering efficiency is designated as the ratio of $\sigma_{sp}$ to $PM_{10}$ concentration. Therefore,
the mass scattering efficiency for $PM_{10}$ aerosols was about 0.67 $m^2g^{-1}$ in the afternoon
and ~0.77 $m^2g^{-1}$ in the morning (~0.25 for heavy dust events, and ~0.70 for the whole
period). Our results were slightly less than ~1.05 $m^2g^{-1}$ in Dunhuang during spring of
2004 (Yan, 2007). Likewise, the mass absorption efficiency was ~0.017 $m^2g^{-1}$ under
heavy dust episodes and ~0.08 $m^2g^{-1}$ in the morning, which was coincident with the
laboratory analytical result of natural desert aerosol at 660 nm (~0.01–0.02 $m^2g^{-1}$) in
Ulan Buh desert (39°26′N, 105°40′E) of northern China (Alfaro et al., 2004). These
diurnal variations of the mass scattering and absorption efficiencies likely reflect the
changes in aerosol chemical composition. The SSA at 670 nm displayed distinct
differences between daytime and nighttime (Figure 4), and the two minimal values at
07:00 LT (~0.90) and 20:00 LT (~0.921) were consistent with the aforementioned
$\sigma_{ap,670}$ diurnal feature. The peak values of SSA (0.945±0.04) for dominant dust
particles in the afternoon agreed well with other field campaigns in Zhangye
(0.95±0.02, Li et al., 2010) and Yulin (0.95±0.04, Xu et al., 2004). The daily low SSA
(0.90–0.92) or overall mean of 0.913±0.055 at Dunhuang was still bigger than that in
both urban (0.81, Bergin et al., 2001) and rural regions (0.81–0.85, Li et al., 2007)
adjacent to Beijing, presumably ascribed to dust particles at night. Yan et al. (2008)



conducted two-year long field measurements at Shangdianzi Global Atmosphere
Watch (GAW) rural site in northern China (~150 km from Beijing) and estimated a
mean SSA of 0.88±0.05, but their data contained summer when aerosol scattering
coefficients may be strengthened by hygroscopic growth and secondary chemical
process.
The wind rose plots give a further insight into the linkages between the
meteorological factors and pollutants, as described in Figure 5. In the morning
(06:00–09:00 LT), a marked northeast wind was prevalent and wind speed was mostly
less than 4 ms$^{-1}$, which revealed that emissions were primarily descended from nearby
farmlands and rural residences (Figure 5a). Although a prominent northwest wind
mainly occurred in the evening hours (19:00–22:00 LT), the east wind and southwest
wind also appeared, which indicated that anthropogenic pollutions came from both
local sources and a relatively large region along the valley (Figure 5b). And Figure 5c
showed the predominant winds were northeast and southwest winds in Dunhuang area,
with the maximal hourly-averaged wind speed exceeding 10 ms$^{-1}$. It was very distinct
that the southwest and northwest winds created higher levels of $PM_{10}$ mass
concentration (>250 $\mu gm^{-3}$), aerosol light scattering coefficient ($\sigma_{sp}$ >150 Mm$^{-1}$) and
absorption coefficient ($\sigma_{ap}$ >8 Mm$^{-1}$), whereas northeast wind generated slightly
smaller concentrations of $PM_{10}$ (~50–100 $\mu gm^{-3}$), $\sigma_{sp}$ (~30–60 Mm$^{-1}$) and $\sigma_{ap}$ (~2–4
Mm$^{-1}$). This was possibly implied that southwest and northwest winds may bring
about dust particles and northeast wind may transport the air pollutants.
**3.3 Local anthropogenic emission sources**
As mentioned above, crop residue burning and agricultural cultivated operations
before the growing season could produce local emission source proximity to the study
area. And sporadic straw burning was indeed to happen throughout the Dunhuang
farmland from 1 April to 10 May 2012, which was the major source of black carbon
surrounding the site. To clarify the potential anthropogenic influence on aerosol
optical properties in desert region, we investigated a typical biomass burning event.
Figure 6 outlines the time series of 5-minute average wind vector (ms$^{-1}$), $PM_{10}$
($\mu gm^{-3}$), $\sigma_{sp}$ at 450, 550, and 700 nm (Mm$^{-1}$), SAE (450–550, 550–700, and 450–700





nm), $\sigma_{ap,670}$ (Mm$^{-1}$), and SSA at 670 nm during a typical Tomb-sweeping Day on 4
April 2012. Tomb-sweeping Day is a Chinese traditional festival for sacrifice rites, in
commemoration of the dead ancestors. To pay homage to loved ones, the people
burned a lot of joss sticks, candles, and paper offerings, and set off firecrackers in that
day throughout the China, which would emit a great amount of air pollutants, such as,
biomass burning aerosol, sulfur dioxide, organic matter, and fugitive dust. From
Figure 6a, slight and variable winds (with wind speed <4 ms$^{-1}$) mainly came from
northeasterly from 00:00 to 12:00 LT, and abruptly changed into weak southeast wind
and south wind, finally, gradually intensified southwest wind (>10 ms$^{-1}$) were
predominant and triggered a severe dust storm from 15:00 LT to the midnight. Prior to
the occurrence of dust episode, the aerosol optical characteristics varied stably, but a
moderate increase was evident during 08:00 to 10:00 LT. For instance, $PM_{10}$
concentration gradually increased from background level ~30 $\mu gm^{-3}$ to a maximum of
62.5 $\mu gm^{-3}$ at about 09:00 LT, $\sigma_{sp,550}$ from ~15 Mm$^{-1}$ to 49.6 Mm$^{-1}$, and $\sigma_{ap,670}$ from
~2.0 Mm$^{-1}$ to 4.75 Mm$^{-1}$. It is ascribed to the contribution of biomass burning in the
process of ritual activities during Tomb-sweeping Day. The SAE value at 450–700 nm
remained invariant (~0.50) before 08:00 LT and sharply rose to a maximal value of
0.87 at 09:00 LT, afterwards gently reduced to around 0.4, which indicated that the
fine-mode particles (i.e., black carbon or soot) were dominated from 08:00 to 10:00
LT. And the SAEs at various wavelengths systematically decreased from 0.4 at 15:00
LT to -0.25 at midnight, suggesting the dust-dominant coarse-mode particles were
prevailed. Meanwhile, the lidar depolarization ratio ($\delta$) also further verified the
existence of small size soot particle. The $\delta$ value preserved steadily at 0.15–0.20
during 08:00 to 10:00 LT, and rapidly attained above 0.3 from 15:00 LT and even
approached 0.50 at intense dust storm (see Fig. 3). The diurnal variation of $SSA_{670}$
showed a more prominent feature, as illustrated in Figure 6f. The $SSA_{670}$ values kept
between 0.88 and 0.92 during 00:00 to 07:00 LT, and dramatically reduced to a
minimum of ~0.83 at 08:30–09:00 LT, then rose to 0.925, confirming the very striking
impacts by light absorbing substances. After 15:00 LT, the $SSA_{670}$ gradually increased
and reached up to about 0.96 during dust storms occurred. Bi et al. (2014) have





demonstrated that dust aerosols shortwave radiative forcing (ARF) at the top of the
atmosphere (TOA) was warming effect when $SSA_{500}$ was less than 0.85, but was
cooling effect when $SSA_{500}$ was greater than 0.85 for Dunhuang Gobi desert area with
high surface albedo. Thereby such significant anthropogenic influence would clearly
modify the microphysical and chemical properties of dust aerosols and eventually
exert remarkable impacts on environmental quality and climatic forcing of dust
particle on both local and regional scales.
**3.4 Dust cases study**
In this section, we particularly explored the absorptive and optical characteristics
of mineral dust during several typical dust cases and discussed its influence on Earth's
radiation balance. Figure 7 provides the wind fields at 500 hPa and 850 hPa levels
during three heavy dust events, based on MERRA reanalysis products. Note that
Dunhuang farmland is marked with a red pentagram and the white areas at 850 hPa
represent the missing values. It is evident that East Asian region was governed by the
powerful and stable westerlies at 500 hPa height on 30 April and 1 May 2012,
whereas two very strong synoptic cyclones at 500 hPa upper atmosphere hovered
about the Mongolia and Kazakhstan respectively on 10 June 2012, matching up with
corresponding cyclone systems appeared at the 850 hPa level. Although there were
missing data in most northwest China, extremely intense northeast wind and east wind
(> 10 ms$^{-1}$) at 850 hPa level were prevailed over the northern territory of Xinjiang
Uygur Autonomous Region during the selected dust storms, where was close to the
Dunhuang site. This could be well confirmed by the simultaneous observations of
wind speed and wind direction nearby the surface at Dunhuang farmland, as
delineated in Figure 8(a). The measured strong northeast and east winds were always
dominated in Dunhuang and 5-min average wind speed attained above 10 ms$^{-1}$ during
intense dust episodes. The selected three dust processes regularly lasted for several
hours during daytime (e.g., from 10:00 to 18:00 LT) and the dust event on 1 April
could be persistent to the midnight, which contributed massive dust particles into the
atmosphere.
There were no measurements of aerosol scattering coefficient ($\sigma_{sp}$) on 10 June due



to equipment failure. From Figure 8, we could know that $PM_{10}$ concentrations usually
exceeded 400 $\mu gm^{-3}$ and even reached up to 1000 $\mu gm^{-3}$ during the heavy dust storms,
and corresponding $\sigma_{sp,550}$ and $\sigma_{ap,670}$ values were generally more than 100 $Mm^{-1}$ and 5
$Mm^{-1}$, respectively, or approached 350 $Mm^{-1}$ and 15 $Mm^{-1}$ in our cases. It is worthy
note that even though pure dust aerosol possesses relatively low light-absorption
ability (with mass absorption efficiency at 660 nm of ~0.01–0.02 $m^2g^{-1}$), the injection
of plentiful mineral particles from dust episodes led to considerably high values of
$\sigma_{ap,670}$. And the SAEs at diverse wavelengths commonly kept at 0.50 or more during
non-dust conditions, while corresponding values dramatically reduced to -0.25~0
under heavy dust cases, which is taken for granted. The $SSA_{670}$ also exhibited
apparent diurnal variations in Figure 8(f). The $SSA_{670}$ values regularly preserved
between 0.88 and 0.92 at nighttime or non-dust weather, and gradually increased to a
maximum of ~0.96–0.98 during strong dust processes, which were close to the
measured value of ~0.97–0.99 for nearly pure Asian dust particles (Anderson et al.,
2003; Bi et al., 2016). These abundant mineral particles in desert source regions were
very likely mixed with local air pollutants especially at night, when the anthropogenic
pollutions favorably built up within the PBL. Moreover, airborne dust particles
ordinarily traveled long distances to downstream areas via mesoscale cyclones, which
would deteriorate the ambient air quality and affected atmospheric chemistry and
climate change on regional scale.

Figure 9 describes the column-integrated aerosol optical depth (AOD) at five

wavelengths (400, 500, 675, 870, and 1018 nm) versus Ångström exponent ($\alpha$) at
400–870 nm on two completely clear-sky days (14 May and 9 June) and two typical
dusty days (30 April and 10 June), which were acquired from sky radiometer (Model
POM-01, PREDE Co. Ltd.). The sky radiometer can measure the direct solar
irradiances and sky diffuse radiances at narrow spectral wavebands during daytime
with 10-minute interval. And the columnar aerosol optical properties under cloudless
conditions were retrieved from sophisticated inversion algorithms (Nakajima et al.,
1996). Note that the cloud contaminated datasets have been eliminated by means of a
series of cloud screening procedures developed by Khatri and Takamura (2009). From





Figure 9, all AOD values under clear-sky days kept very stable variations throughout
the day and ranged from 0.02 to 0.12, which were comparable to the clean
background levels in the central Tibetan Plateau (Xia et al., 2011) and Badain Jaran
Desert (Bi et al., 2013). And the corresponding Ångström exponent α on 14 May and
9 June were greater than 0.6, indicating extremely low aerosol loading. In contrast,
the AODs under dust events (30 April and 10 June) displayed pronounced diurnal
variations and all AOD values were larger than 0.30 (with maximum of 0.60), and α
varied between 0.10 and 0.25, representing high dust concentration levels. These
elevated dust particles in the atmosphere would readjust the energy distributions of
solar radiative fluxes at the surface.

Based on aforementioned measurements of total sky imager, micro-pulse lidar and

sky radiometer, we identified three completely clear-sky days (14 May, 29 May, and 9
June) and two "clean" dusty days (30 April and 10 June). The "clean" dusty days in
this study were denoted as the dust storms weather without the influence of clouds.
This afforded us a good opportunity to elucidate the potential impacts of dust events
on radiation balance at the ground. Figure 10 draws the 1-minute average solar direct
normal radiation, sky diffuse radiation, total shortwave radiation, and downward
long-wave radiation fluxes under the selected five days, which were derived from the
high-precision broadband radiometers as described in section 2.3. All radiative
quantities presented smooth diurnal variations under clear-sky cases (14 May, 29 May,
and 9 June). The airborne dust particles impeded the sunlight to the ground through
scattering and absorbing solar radiation, for instance, they could significantly reduced
the surface direct radiative fluxes in daytime about 200–350 Wm$^{-2}$ (Figure 10a),
whereas considerably increased the surface diffuse radiative fluxes up to ~150–300
Wm$^{-2}$ (Figure 10b). As a result, the overall attenuation effect on total shortwave
radiative fluxes varied between –150 and –50 Wm$^{-2}$. The incoming solar energy
absorbed by dust particles would heat the atmospheric dust layer (Bi et al., 2014) that
likely played a profound role in atmospheric boundary layer structure and cloud
microphysical process (J. Huang et al., 2006, 2010; Li et al., 2016). The downward
longwave radiation (DLW) at the surface was majorly reliant on the clouds, water



vapor, $CO_2$, and other greenhouse gases (Wang and Dickinson, 2013). In general, the
presence of clouds in the atmosphere would fluctuate drastically the diurnal variation
of DLW. And the smooth changes of DLW under both clear-sky and dusty days in
Figure 10d revealed the robustness of the cloud screening method used in this paper.
Figure 10d displays that the DLW values under dusty cases were always greater than
that in clear-sky cases, with the total average differences of +40~+60 $Wm^{-2}$. The
warming dust layer could enhance the surface DLW, hence the dust particles should
contribute a few percentages to the increased DLW, but not all. This is because the
potential greenhouse gases in the atmosphere could substantially affect the DLW
variations. For instance, the DLW on 9 June were distinctly lager than that in other
cloudless cases (i.e., 14 and 29 May) and the dusty case of 30 April. It is partly
attributable to the higher RH values on 9 June than that in other days.
**4. Concluding remarks**
In this article, we surveyed the optical features and size distribution of dust
aerosol in a Gobi farmland region of northwest China from 1 April to 12 June 2012,
and uncovered a potential anthropogenic influence. The overall average $PM_{10}$ mass
concentration, light scattering coefficient ($\sigma_{sp,670}$), absorption coefficient ($\sigma_{ap,670}$), and
single-scattering albedo ($SSA_{670}$) throughout the experiment were $113\pm169$ $\mu gm^{-3}$,
$53.3\pm74.8$ $Mm^{-1}$, $3.2\pm2.4$ $Mm^{-1}$, and $0.913\pm0.05$, which were comparable to the
background levels in southern United States, but lower than that in the eastern and
other northwestern China. Frequent dust storms could markedly elevate dust loading
and dominated the temporal evolution of airborne aerosol in Dunhuang region. The
hourly average $PM_{10}$, $\sigma_{sp,670}$, and $\sigma_{ap,670}$ reached up to 2000 $\mu gm^{-3}$, 800 $Mm^{-1}$, and 25
$Mm^{-1}$ during the severe dust events that were tenfold greater than the total mean
values, along with the particle size concentrated in diameters of 1–3 $\mu m$. Meanwhile,
the correspondingly high $SSA_{670}$ (~0.96–0.98) and depolarization ratio ($\delta$ of ~0.3–0.5),
and low SAE (-0.25~0) values adequately verified the presence of coarse-mode
mineral dust, resulting in significantly reducing the solar direct radiation (~200–350
$Wm^{-2}$) and increasing diffuse radiation (~150–300 $Wm^{-2}$) at the surface, and hence

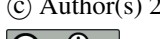



affecting the regional climate.
Owing to relatively low aerosol levels observed in Dunhuang, any slightly
anthropogenic perturbation would induce a substantial influence on the aerosol
physicochemical property. The so-called anthropogenic dust produced by agricultural
cultivating operations (e.g., land planning, plowing, and disking) brought a significant
superimposed effect on high dust concentrations in Dunhuang farmland prior to the
growing season, when the underlying surface was primarily covered with bare soils.
This to some extent could be interpreted the drastic changes of aerosol loadings in
April and early May. In contrast, the local pollutant emissions mainly affected the
absorptive characteristics of dust aerosol especially at night, when the anthropogenic
pollutants favorably accumulated within the PBL and likely mixed with abundant
mineral dust in the atmosphere. Therefore, the diurnal variations of $\sigma_{ap,670}$ and $SSA_{670}$
exhibited prominent features, both of which have got two peak values at night and in
the early morning. For instance, ~3.3 $Mm^{-1}$ at 20:00 LT and ~2.9 $Mm^{-1}$ at 08:00 LT
for $\sigma_{ap,670}$ were much more than the low level of ~2.0 $Mm^{-1}$ in the afternoon, which
was attributed to the influence of anthropogenic emissions. And the mean $SSA_{670}$ of
predominant dust particles in the afternoon (13:00–18:00 LT) was 0.945±0.04 that
was evidently greater than the mixed dust-pollutants dominated $SSA_{670}$ of ~0.90 at
07:00 LT and ~0.92 at 20:00 LT.
The findings of this study directly demonstrated mineral dust in Dunhuang
farmland was substantially affected by anthropogenic pollutants, which would help to
promote a further insight into the interaction among dust aerosol, atmospheric
chemistry, and regional climate in desert source region. However, the potentially
anthropogenic influences on dust aerosol in Dunhuang showed far smaller than that
measured in eastern China, which was expected for the remote desert areas with
sparsely population and lesser human activities. Recently, Huang et al. (2016)
indicated that most of the drylands in the world were fragile and susceptible to climate
change and human activities and would be subject to the acceleration of drought
expansion by the end of twenty-first century. Under the possible scenario, it is very
critical to make clear the relative contributions of natural and anthropogenic forcing



factors on global climate change, such as, natural dust and anthropogenic dust, which
calls for further investigating through a lot more observations and technologies.
**5. Data availability**
All ground-based aerosol datasets used in this paper are available via contacting
Jianrong Bi (bijr@lzu.edu.cn).

*Acknowledgements*. This work was jointly supported by the Foundation for Innovative
Research Groups of the National Natural Science Foundation of China (41521004), the National
Natural Science Foundation of China (41575015 and 41405113), the Fundamental Research Funds
for the Central Universities lzujbky-2015-4 and lzujbky-2016-k01, and the China 111 Project (No.
B 13045). The authors would like to express special thanks to David S. Covert for guiding the
in-situ aerosol measurements. We thank the OMI and MERRA teams for supplying the satellite
data and reanalysis products used in this study. We also appreciate all anonymous reviewers for
their constructive and insightful comments.

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






**Figure captions**



**Table 1.** The main aerosol observing and other ground-based instruments deployed at Dunhuang
farmland during spring of 2012.

| Measured variables | Model, Manufacturer | Accuracy |
|---|---|---|
| $PM_{10}$ concentration | Ambient particulate monitor (RP1400a), R&P Corp. | 0.1 $\mu gm^{-3}$ |
| Aerosol scattering coefficient | Integrating nephelometer (TSI 3563), TSI Inc. 450, 550, and 700 nm | 0.44, 0.17, and 0.26 $Mm^{-1}$ |
| Aerosol absorption coefficient | Multi-angle absorption photometer (MAAP 5012), Thermo | 0.66 $Mm^{-1}$ |
| Aerosol size distribution | Aerodynamic particle sizer (APS 3321), TSI Inc., 0.5~20 μm | 0.001 $cm^{-3}$ |
| Aerosol-attenuated backscatter profile | Micro-pulse lidar (MPL–4), Sigma Space Corp. | Spatial resolution: ~30 m |
| Meteorological elements | Weather transmitter (WXT–520), Vaisala, Ta, RH, P, u, WD | Ta: ±0.3℃, RH: 0.1%, P: 0.1 hPa, u: 0.1 $ms^{-1}$, WD:1° |
| Global and diffuse radiation | Pyranometer (PSP[a,b]), Eppley Lab., 0.285~2.8 μm | Global: 8.46, diffuse: 8.48 $\mu VW^{-1}m^{-2}$ |
| Direct radiation | Pyrheliometer (NIP[b]), Eppley Lab., 0.285~2.8 μm | 8.38 $\mu VW^{-1}m^{-2}$ |
| Downward long wave radiation | Pyrgeometer (PIR[a,b]), Eppley Lab., 3.5~50 μm | 2.98 $\mu VW^{-1}m^{-2}$ |
| 24–bit color JPEG image | Total Sky Imager (TSI880), YES Inc., 352×288 pixel | Sampling rate: 1 minute |

[a]The instrument is equipped with the Eppley ventilation system (VEN).
[b]The instrument is mounted on a two-axis automatic sun tracker (Model 2AP, Kipp&Zonen).


**Table 2.** Statistical summary of hourly average aerosol optical properties measured during
intensive observation period[a]

| Variable | Mean | Std[b] | Median | 10th percentile | 25th percentile | 75th percentile | 90th percentile |
|---|---|---|---|---|---|---|---|
| $PM_{10}$ ($\mu gm^{-3}$) | 113 | 169 | 54 | 17 | 29 | 111 | 300 |
| $\sigma_{sp}$ ($Mm^{-1}$) | 53.3 | 74.8 | 28.3 | 11.2 | 16.0 | 55.8 | 123.5 |
| $\sigma_{ap}$ ($Mm^{-1}$) | 3.20 | 2.40 | 2.50 | 1.27 | 1.69 | 3.90 | 5.94 |
| SSA (670 nm) | 0.913 | 0.055 | 0.923 | 0.850 | 0.892 | 0.949 | 0.967 |
| SAE (450/700 nm) | 0.45 | 0.45 | 0.42 | −0.1 | 0.1 | 0.73 | 0.99 |

[a]All aerosol data reported for volumes under 1013.25 hPa and 20 ℃.
[b]Std denotes the standard deviation.









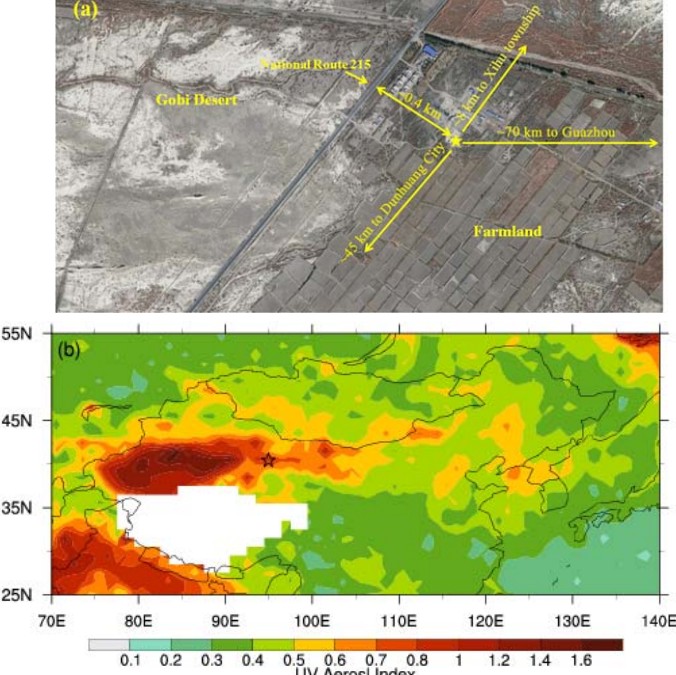



**Figure 1.** (a) The Dunhuang farmland site (40.492°N, 94.955°E, altitude: 1061 m) labeled with a
pentagram and its surrounding region. (b) OMI (Ozone Monitoring Instrument, 2004) mean UV
aerosol index from 1 April to 12 June 2012. The site is located in the downwind region of the
Taklimakan Desert and frequently outbreaks dust storms.


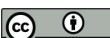


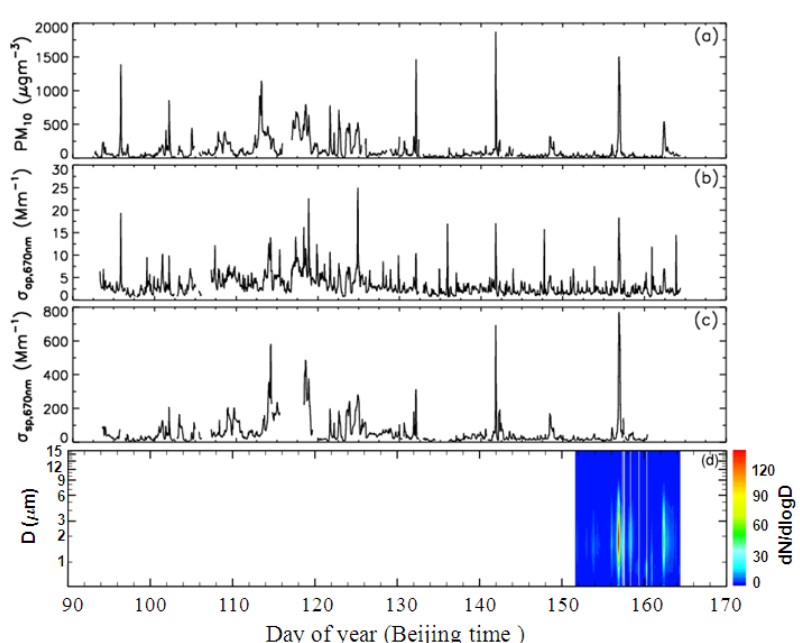


**Figure 2.** Time series of hourly average (a) $PM_{10}$ mass concentration in $\mu gm^{-3}$, (b) aerosol
absorption coefficient at 670 nm, (c) aerosol scattering coefficient at 670 nm, and (d) aerosol size
distribution in $cm^{-3}$ at Dunhuang farmland during the whole sampling period.



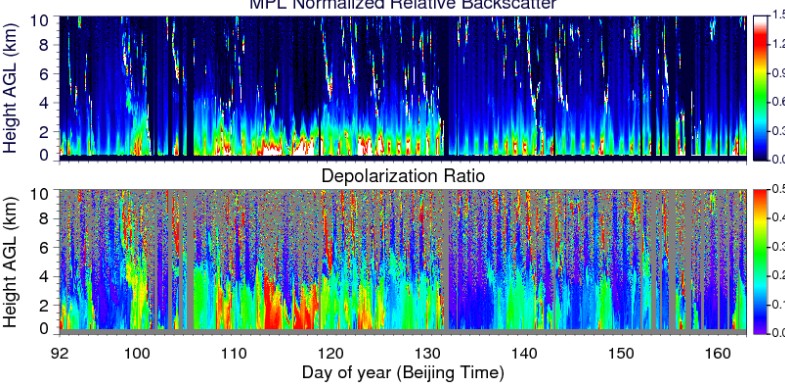


**Figure 3.** Time evolutions of the MPL normalized relative backscatter intensity (top panel) and
depolarization ratio (bottom panel) at Dunhuang farmland from 1 April to 12 June 2012.










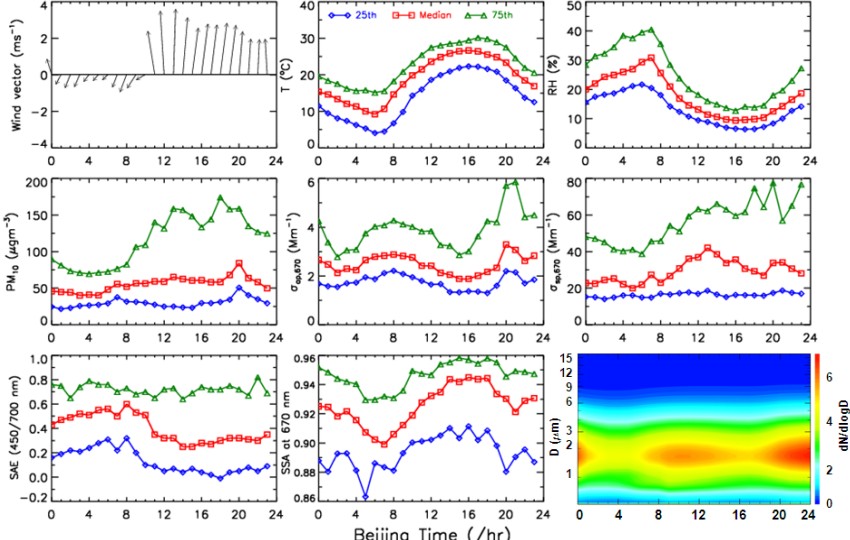


**Figure 4.** The diurnal variations of (first row, left to right) wind vector (ms$^{-1}$), air temperature (T in ℃), relative humidity (RH in %), (second row, left to right) PM$_{10}$ concentration (μgm$^{-3}$), aerosol scattering coefficient at 670 nm ($\sigma_{sp,670}$ in Mm$^{-1}$), aerosol absorption coefficient at 670 nm ($\sigma_{ap,670}$ in Mm$^{-1}$), (third row, left to right) scattering Ångström exponent at 450–700 nm (SAE 450/700 nm), aerosol single-scattering albedo at 670 nm (SSA$_{670}$), and aerosol size distribution (dN/dlogD in cm$^{-3}$) in Dunhuang site from 1 April to 12 June 2012 (30 May to 12 June for aerosol size distribution). Median values (red square) are shown to give a more apparent diurnal feature than mean values, which could be affected by several strong dust episodes. The 25[th] (blue diamond) and 75[th] (green triangle) percentiles for each hour of the day are also displayed.





**Figure 5.** Wind rose plots for (a) morning hour (06:00–09:00 LT), (b) evening hour (19:00–22:00
LT), and (c) all hours; shade represents wind speed (ms$^{-1}$). Wind roses for all hours, with shade
representing levels of (d) PM$_{10}$ concentration (μgm$^{-3}$), (e) aerosol scattering coefficient at 670 nm
($\sigma_{sp}$ in Mm$^{-1}$), and (f) aerosol absorption coefficient at 670 nm ($\sigma_{ap}$ in Mm$^{-1}$).






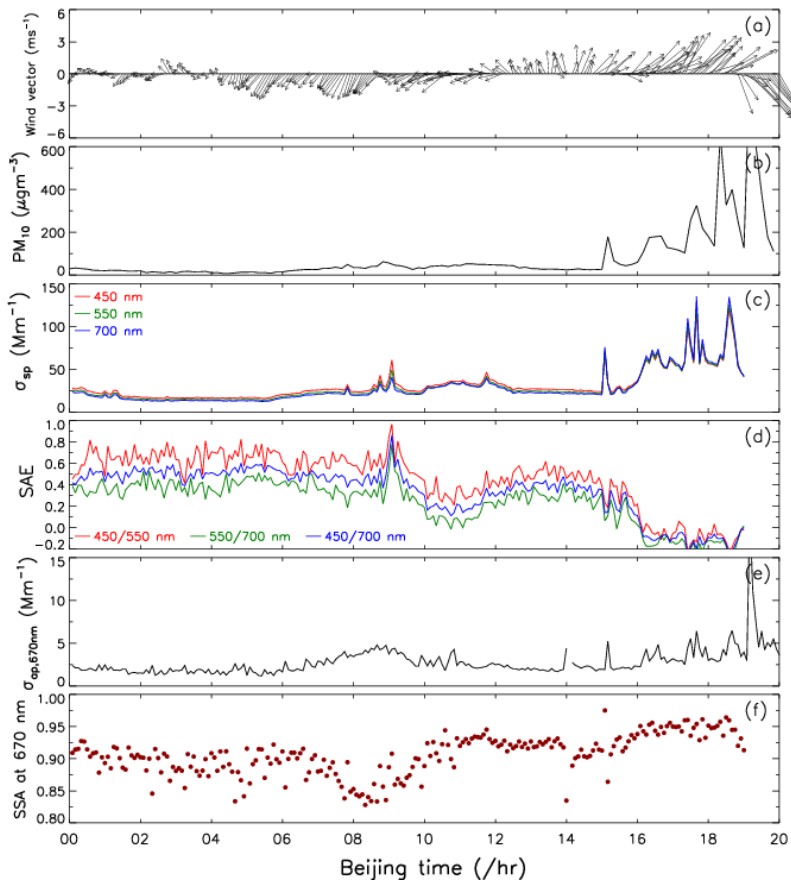


**Figure 6.** Time series of (a) wind vector (ms$^{-1}$), (b) PM$_{10}$ concentration (μgm$^{-3}$), (c) aerosol
scattering coefficient (σ$_{sp}$ in Mm$^{-1}$) at 450 nm (red), 550 nm (green), and 700 nm (blue), (d)
scattering Ångström exponent (SAE) at 450–550 nm (red), 550–700 nm (green), and 450–700 nm
(blue), (e) aerosol absorption coefficient at 670 nm (σ$_{ap}$ in Mm$^{-1}$), and (f) single-albedo albedo at
670 nm (SSA$_{670}$) during a typical Tomb-sweeping Day on 4 April 2012, which implies a potential
anthropogenic influence on aerosol optical properties. All data points are obtained from 5-minute
average values.




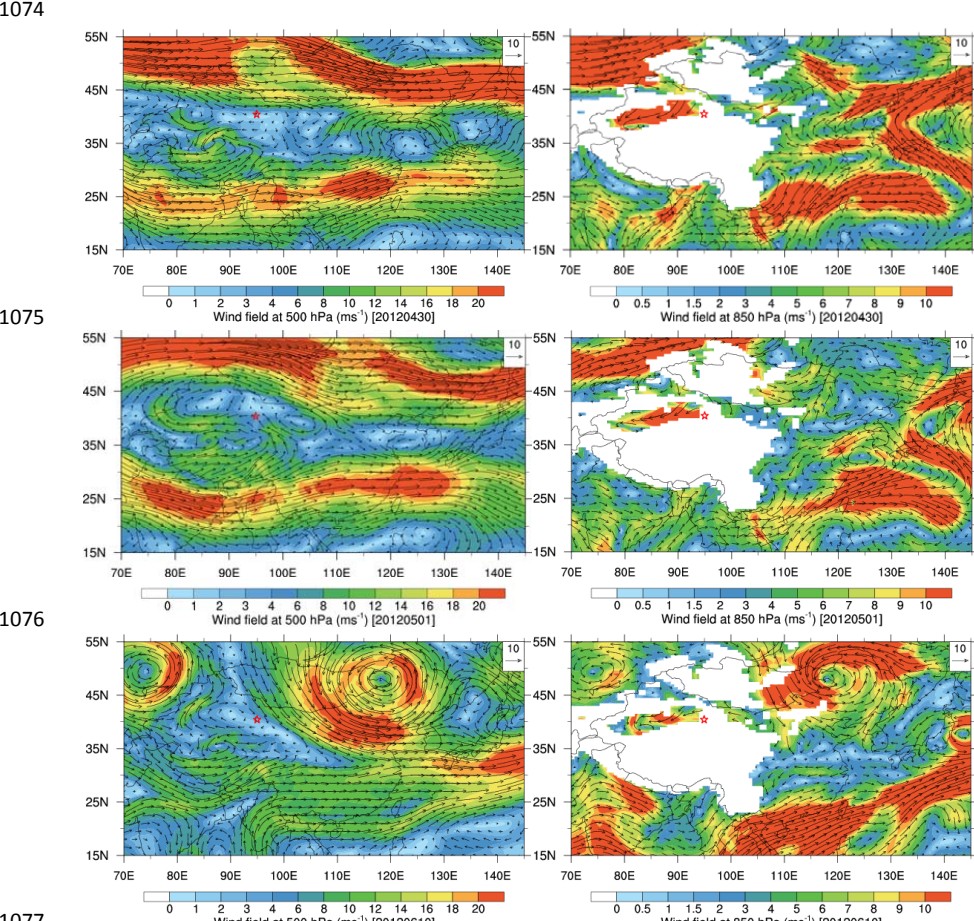


**Figure 7.** The wind fields (black arrows) at 500 hPa (left panel) and 850 hPa (right panel) levels during three heavy dust events on 30 April (top), 1 May (middle), and 10 June (bottom) 2012, based on MERRA reanalysis data. Note that the Dunhuang farmland is marked with a red pentagram and the white regions at 850 hPa are on behalf of the missing values.





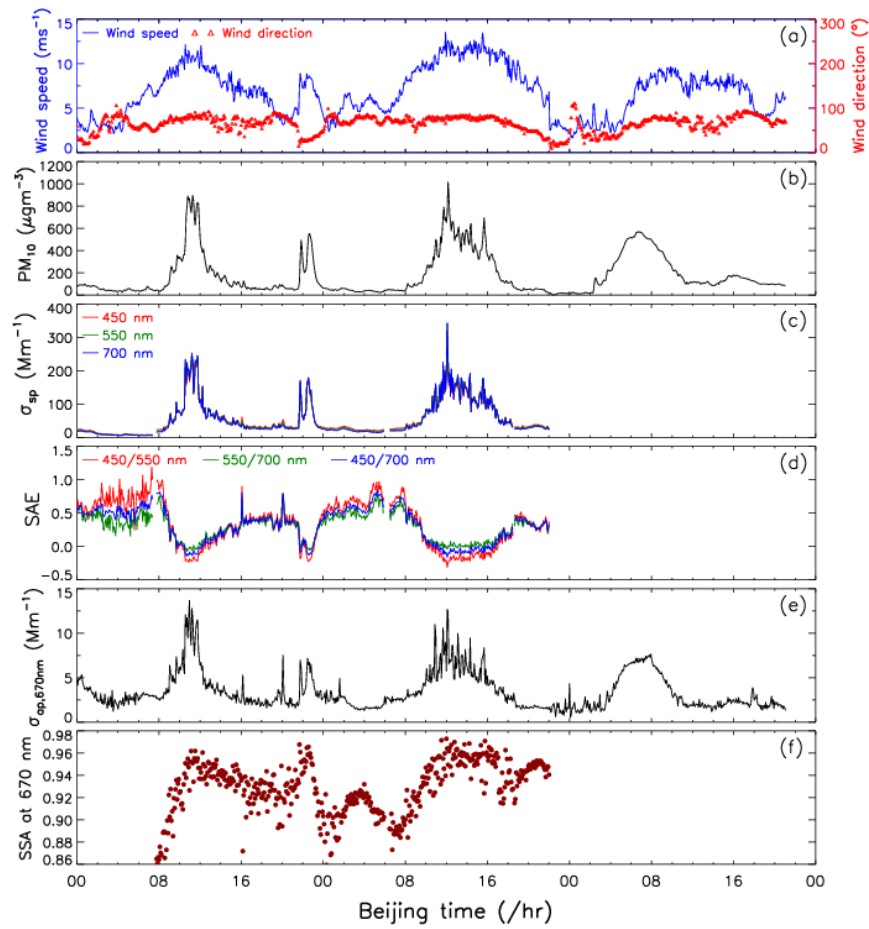


**Figure 8.** The same as Figure 6, except for (a) wind speed (ms$^{-1}$) and wind direction (°) during
three heavy dust events on 30 April, 1 May, and 10 June 2012. There were no measurements of
aerosol scattering coefficient ($\sigma_{sp}$ in Mm$^{-1}$) on 10 June due to equipment failure.




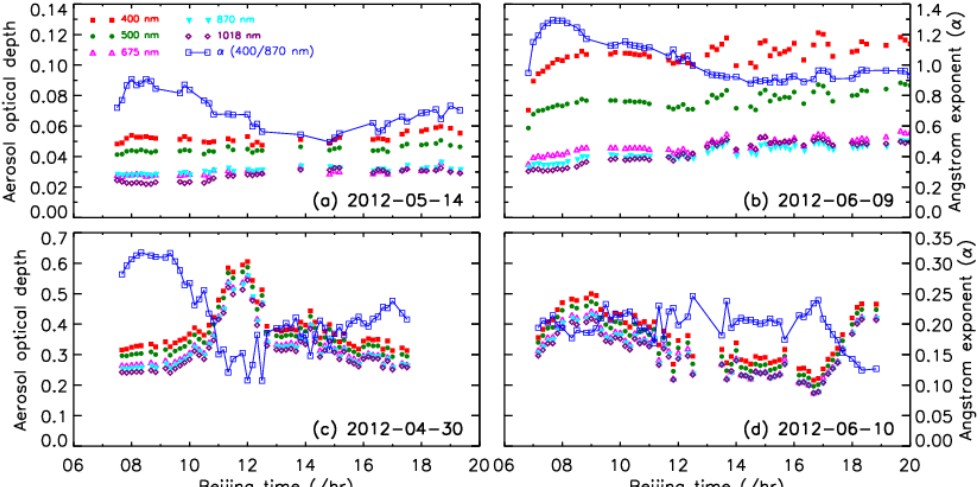


**Figure 9.** Time evolutions of aerosol optical depth (AOD) at five wavelengths (400, 500, 675, 870,
and 1018 nm) versus Ångström exponent ($\alpha$) at 400–870 nm on (a) 14 May, (b) 9 June, (c) 30
April, and (d) 10 June 2012. Note that Figures 9(a)–9(b) are adopted from *Bi et al.* (2014) with an
addition of the Ångström exponent plot in the original publication.


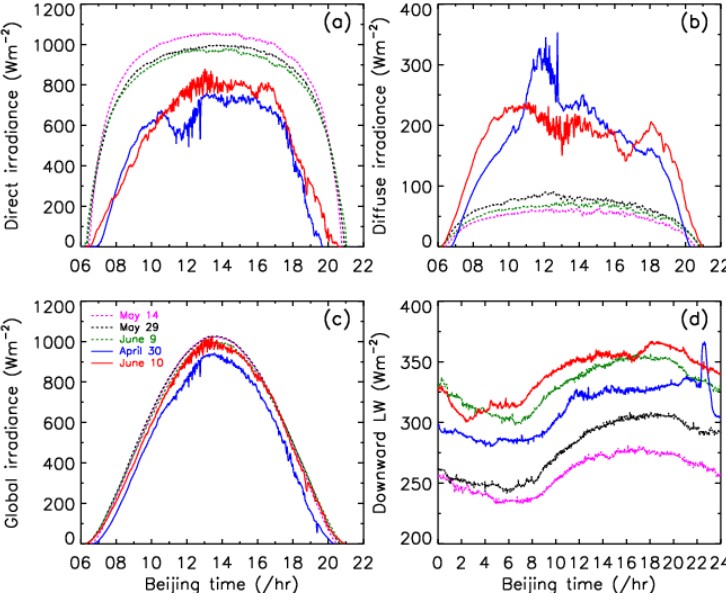


**Figure 10.** Diurnal variations of ground-based measured of 1-minute average (a) direct, (b) diffuse,
and (c) global irradiances, and (d) downward long wave irradiance under completely clear–sky
conditions (14 May, 29 May, and 9 June) and dust events (30 April and 10 June).
