# Peer review of "Measurement of scattering and absorption properties of dust aerosol"

_Atmospheric Chemistry and Physics, 2017_

## Referee Comment (RC1) · Anonymous Referee #1 · 17 Mar 2017

**General comments:**

Dust aerosol in remote Taklimakan Desert and Gobi deserts of northwest China is thought to be hardly affected by human activities, due to sparsely population. The authors conducted a comprehensive field measurement in a Gobi farmland region of northwest China, and demonstrated a potential anthropogenic influence on dust physicochemical properties using multiple ground-based active and passive sensors. The agricultural operations and biomass burning from crop residue prior to growing season were well documented to produce significant impacts on elevated dust loadings and absorption characteristics in Dunhuang farmland during spring of 2012. The findings of this study are very interesting and would help to improve our understanding of the interaction among dust aerosol, atmospheric chemistry, and climate change in desert source region. And I suggest that the authors should carry out long-term and continuous measurements of mineral dust at remote Gobi deserts in northwest China, to quantify the potential anthropogenic contributions on regional climatic and environmental changes. I think the English wring is fine, and I recommend this manuscript is appropriate for publishing after minor revision.

**Minor comments:**

1. **Abstract**, Page 1, line 27: "In the afternoon (13:00–18:00 LT)"

⇒ Change to "In the afternoon (13:00–18:00 LT, local time)". When an abbreviation firstly appears in the manuscript, please give the full name.

2. Page 3, line 78: "(i.e., hematite and goethite)"

⇒ Change to "(i.e. hematite and goethite)"

3. Page 4, line 90: "(i.e., Mongolia Gobi desert)"

⇒ Change to "(i.e. Inner Mongolian Gobi desert)"

4. Page 4, line 111: "close to the east edge of Kumtag Desert"

⇒ Change to "close to the eastern edge of Kumtag Desert"

5. Page 5, line 130: "to the southeast"

⇒ Change to "to the southwest"

6. Page 6, line 154: "High AI values (>0.7) distributions"

⇒ Change to "The distributions of high AI values (>0.7)"

7. Page 9, line 265: "(i.e., Mongolia cyclones)"

⇒ Change to "(i.e. Mongolian cyclone)"

8. Page 12, line 331: "2 to 4 km"

⇒ Change to "4 km"

9. Page 12, line 332: "which was within the planetary boundary layer (PBL)"

⇒ Change to "which was above the planetary boundary layer (PBL)"

10. Page 14, line 402: "Likewise"

⇒ Change to "Similarly"

11. Page 19, line 563: "atmospheric boundary layer structure"

⇒ Change to "the structure of atmospheric boundary layer"

12. Page 20, line 575: "lager"

⇒ Change to "larger"

13. Page 21, line 614: "The findings of this study directly demonstrated mineral dust"

⇒ Change to "The findings of this study directly demonstrated that mineral dust"

---

## Author Comment (AC1) · 8 Apr 2017

Response to Referee1:

We are grateful to the Editor's and Referee1's insightful and constructive comments for this manuscript! We have carefully checked and revised the whole manuscript according to Referee1's comments, which are helpful and valuable for greatly improving our manuscript. Please find a point-by-point reply to the issues as follows (highlighted in the blue font). And we have also uploaded the file of "Response to-Referee1(acp-2017-165).pdf".

[Figure]

General comments:

Dust aerosol in remote Taklimakan Desert and Gobi deserts of northwest China is thought to be hardly affected by human activities, due to sparse population. The authors conducted a comprehensive field measurement in a Gobi farmland region of northwest China, and demonstrated a potential anthropogenic influence on dust physicochemical properties using multiple ground-based active and passive sensors. The agricultural operations and biomass burning from crop residue prior to growing season were well documented to produce significant impacts on elevated dust loadings and absorption characteristics in Dunhuang farmland during spring of 2012. The findings of this study are very interesting and would help to improve our understanding of the interaction among dust aerosol, atmospheric chemistry, and climate change in desert source region. And I suggest that the authors should carry out long-term and continuous measurements of mineral dust at remote Gobi deserts in northwest China, to quantify the potential anthropogenic contributions on regional climatic and environmental changes. I think the English wring is fine, and I recommend this manuscript is appropriate for publishing after minor revision.

Response: Thank you very much for the Referee's good suggestions and the acceptance of this work. Indeed, this study only covers several months in spring during intensive period and it is indispensable to acquire long-term measurements of mineral dust for fully understanding the potential anthropogenic contributions on regional environmental and climatic changes. Hence, we have set up two permanent field observatories (SACOL and Dunhuang) in northwest China to continuously measure mineral dust since 2013, and will obtain more valuable findings, which will help quantify the anthropogenic contributions of dust aerosol in remote desert source region.

Minor comments:

1. Abstract, Page 1, line 27: "In the afternoon (13:00–18:00 LT)"

âĞŠ Change to "In the afternoon (13:00–18:00 LT, local time)". When an abbreviation

firstly appears in the manuscript, please give the full name.

Response: We have changed "In the afternoon (13:00–18:00 LT)" to "In the afternoon (13:00–18:00 LT, local time)" in Line 27 and modified the corresponding places in the entire context.

2. Page 3, line 78: "(i.e., hematite and goethite)"

âĞŠ Change to "(i.e. hematite and goethite)"

Response: We have changed to "(i.e. hematite and goethite)" in Line 78.

3. Page 4, line 90: "(i.e., Mongolia Gobi desert)"

âĞŠ Change to "(i.e. Inner Mongolian Gobi desert)"

Response: We have changed to "(i.e. Inner Mongolian Gobi desert)" in Line 90.

4. Page 4, line 111: "close to the east edge of Kumtag Desert"

âĞŠ Change to "close to the eastern edge of Kumtag Desert"

Response: We have changed to "close to the eastern edge of Kumtag Desert" in Line 111.

5. Page 5, line 130: "to the southeast"

âĞŠ Change to "to the southwest"

Response: We have changed to "to the southwest" in Line 130.

6. Page 6, line 154: "High AI values (>0.7) distributions"

âĞŠ Change to "The distributions of high AI values (>0.7)"

Response: We have changed "High AI values (>0.7) distributions" to "The distributions of high AI values (>0.7)" in Line 154.

7. Page 9, line 265: "(i.e., Mongolia cyclones)"

âĞŠ Change to "(i.e. Mongolian cyclone)"

Response: We have changed "(i.e., Mongolia cyclones)" to "(i.e. Mongolian cyclone)" in Line 265.

8. Page 12, line 331: "2 to 4 km"

âĞŠ Change to "4 km"

Response: We have changed "2 to 4 km" to "4 km" in Line 331.

9. Page 12, line 332: "which was within the planetary boundary layer (PBL)"

âĞŠ Change to "which was above the planetary boundary layer (PBL)"

Response: We have changed to "which was above the planetary boundary layer (PBL)" in Line 332.

10. Page 14, line 402: "Likewise"

âĞŠ Change to "Similarly"

Response: We have changed "Likewise" to "Similarly" in Line 402.

11. Page 19, line 563: "atmospheric boundary layer structure"

âĞŠ Change to "the structure of atmospheric boundary layer"

Response: We have changed "atmospheric boundary layer structure" to "the structure of atmospheric boundary layer" in Line 563.

12. Page 20, line 575: "lager"

âĞŠ Change to "larger"

Response: We have changed "lager" to "larger" in Line 575.

13. Page 21, line 614: "The findings of this study directly demonstrated mineral dust"

[Figure]

âĞŠ Change to "The findings of this study directly demonstrated that mineral dust"

Response: We have changed to "The findings of this study directly demonstrated that mineral dust" in Line 614

Please also note the supplement to this comment:
http://www.atmos-chem-phys-discuss.net/acp-2017-165/acp-2017-165-AC1-supplement.pdf

---

## Referee Comment (RC2) · Anonymous Referee #2 · 3 May 2017

**Comment on "Measurement of scattering and absorption properties of dust aerosol in a Gobi farmland region of northwest China—a potential anthropogenic influence" by Bi et al.**

This manuscript presents the measurement of scattering and absorption properties of dust aerosol from a comprehensive field campaign in a Gobi farmland region of northwest China during spring 2012. Overall, the manuscript could make a good contribution to the scientific research by providing useful scientific knowledge on the interaction among dust aerosol, atmospheric chemistry, and climate change in desert source region.. However, I believe that the manuscript needs the following minor revisions before it is accepted for publication by ACP:.

1) Lines 22-24: Please present the more results and discussions on the statement in the text about the statement in the abstract that "The anthropogenic dust produced by agricultural cultivations (e.g., land planning, plowing, and disking) exerted a significant superimposed effect on high dust concentrations in Dunhuang farmland prior to the growing season (i.e., from 1 April to 10 May)."

2) Lines 25-27: It is a misleading conclusion that "Strong south valley wind and vertical mixing in daytime scavenged the pollution and weak northeast mountain wind and stable inversion layer at night favorably accumulated the air pollutants near the surface." Please follow the diurnal changes of winds and $PM_{10}$ in Figs. 4 and 6.

3) It could be unnecessary to present the wind fields at 500 hPa and 850hPa levels from the MERRA reanalysis products in Fig.7, because the dust aerosols in a Gobi farmland region of northwest China are mostly the local emissions and a short-distance transport to the measurement site within the boundary layer.

4) Line 532: "mesoscale cyclones" should be "synoptic cyclones".

5) Lines 570-577: It is an interesting result that Figure 10d displays that the DLW values under dusty cases were always greater than that in clear-sky cases, with the total average differences of +40~+60 Wm-2." . However, the interpretation is unconvinced. From Fig. 10d, it could be seen that the warming dust layer could enhance the surface DLW with a large (+40~+60 Wm-2.:not a few percentages!) contribution to the increased DLW. It is unreasonable that the

potential greenhouse gases in the atmosphere could substantially affect the DLW differences between dusty and clear-sky cases (Fig. 10d). Also, please present the measured cloud cover or RH on April 9 to support the statement that "it is partly attributable to the higher RH values on 9 June than that in other days".

6) Please improve the quality of all the Figs., with clarifying the figure captions, such as horizontal wind vector in Figs 4, near surface wind in Figs. 6 and 8, and the same color curves for all the Figs. 10a, 10b,10c and 10d,

---

## Author Comment (AC2) · 19 May 2017

Response to Referee-2:

We appreciate the Editor and Referee-2's valuable and constructive comments for this manuscript, which greatly assist in improving the quality of the original manuscript! We have carefully checked and revised the whole manuscript according to Referee-2's comments. Please find a point-by-point reply to the issues as follows. And we have also uploaded the file of 'acp-2017-165-supplement.pdf'.

General comments:
This manuscript presents the measurement of scattering and absorption properties of dust aerosol from a comprehensive field campaign in a Gobi farmland region of northwest China during spring 2012. Overall, the manuscript could make a good contribution to the scientific research by providing useful scientific knowledge on the interaction among dust aerosol, atmospheric chemistry, and climate change in desert source region. However, I believe that the manuscript needs the following minor revisions before it is accepted for publication by ACP.

Response: Thank you very much for the Referee's insightful suggestions and constructive comments on this manuscript. We have carefully checked and revised the whole manuscript according to Referee2's comments. Please find a point-by-point reply to the issues as follows.

Minor comments:

1. Lines 22–24: Please present the more results and discussions on the statement in the text about the statement in the abstract that "The anthropogenic dust produced by agricultural cultivations (e.g., land planning, plowing, and disking) exerted a significant superimposed effect on high dust concentrations in Dunhuang farmland prior to the growing season (i.e., from 1 April to May)."

Response: We have presented visual photos of a variety of agricultural cultivations in Dunhuang farmland (nearby SACOL's Mobile Facility) prior to the growing season (i.e. from 1 April to 10 May, 2012), as shown in Figure S1. Diverse agricultural operations (e.g., land planning, plowing, disking and laying plastic mulch) were carried out in loose and bare Dunhuang farmland from 1 April to 10 May, 2012, which produced massive soil dust into the atmosphere, especially under strong surface winds (see Figure S1a-c). Therefore, the mass concentrations of particulate matter (PM10) in the source and adjacent downwind regions (including SACOL's Mobile Facility) were significantly elevated by these human activities. In contrast, the crops in Dunhuang farmland gradually become green since 10 May, 2012, indicating the coming of growth season (Figure

[Figure]

S1f).

We also added more discussions about this in the context (Page 10, Lines 272–278). Please check our revised manuscript in detailed.

2. Lines 25–27: It is a misleading conclusion that "Strong south valley wind and vertical mixing in daytime scavenged the pollution and weak northeast mountain wind and stable inversion layer at night favorably accumulated the air pollutants near the surface." Please follow the diurnal changes of winds and PM10 in Figs. 4 and 6.

Response: Thank you very much for your suggestions! The conclusion here corresponds to the diurnal changes of winds and PM10 in Figs. 4 and 6, which can be interpreted by classical mountain-valley wind circulation. Please refer to more detailed explanations in Pages 12–13, Lines 336–359.

3. It could be unnecessary to present the wind fields at 500 hPa and 850 hPa levels from the MERRA reanalysis products in Fig. 7, because the dust aerosols in a Gobi farmland region of northwest China are mostly the local emissions and a short-distance transport to the measurement site within the boundary layer.

Response: Thank you very much for your good comments! We mainly intend to elucidate two points using Fig. 7. Firstly, the selected three heavy dust events (i.e. 30 April, 1 May, and 10 June) were triggered by different synoptic cyclones. East Asian region was governed by the powerful and stable westerlies at 500 hPa height on 30 April and 1 May 2012, whereas strong Mongolian cyclones at 500 hPa upper atmosphere hovered about the southern Mongolia on 10 June 2012. Secondly, the ground-based measured strong northeast and east winds (> 10 ms-1) under three dust events were completely consistent with the wind fields at 850 hPa levels from the MERRA reanalysis products, which indicated the studied dust events were regional scales instead of local scales.

4. Line 532: "mesoscale cyclones" should be "synoptic cyclones".

Response: We have changed "mesoscale cyclones" to "synoptic cyclones" in Line 532.
5. Lines 570–577: It is an interesting result that Figure 10d displays that "the DLW values under dusty cases were always greater than that in clear-sky cases, with the total average differences of +40 +60 Wm-2". However, the interpretation is unconvinced. From Fig. 10d, it could be seen that the warming dust layer could enhance that surface DLW with a large (+40 +60 Wm-2: not a few percentages!) contribution to the increased DLW. It is unreasonable that the potential greenhouse gases in the atmosphere could substantially affect the DLW differences between dusty and clear-sky cases (Fig. 10d). Also, please present the measured cloud cover or RH on April 9 to support the statement that "it is partly attributable to the higher RH values on 9 June than that in other days."

Response: Thank you very much for your insightful and valuable comments! Indeed, the warming dust layer could enhance that surface DLW with a large contribution to the increased DLW (+40 +60 Wm-2: not a few percentages!). Hence, we have changed "contribute a few percentages to the increased DLW" to "contribute a large percentages to the increased DLW" in Line 576.

"the potential greenhouse gases in the atmosphere could substantially affect the DLW variations.": "the greenhouse gases" in the manuscript represent the presence of water vapor or clouds in the atmosphere, which causes confusion. Therefore, we have changed "This is because the potential greenhouse gases in the atmosphere could substantially affect the DLW variations." to "This is because the potential water vapor in the atmosphere could substantially affect the DLW variations." in Lines 573-575. Meanwhile, we have presented the diurnal variations of 10-second average relative humidity (RH,

6. Please improve the quality of all the Figs., with clarifying the figure captions, such as horizontal wind vector in Figs. 4, near surface wind in Figs. 6 and 8, and the same color curves for all the Figs. 10a, 10b, 10c and 10d.

Response: Thank you very much for your valuable comments for improving the quality

of this manuscript! We have improved the quality of all the Figs. in the context and corrected the same color curves for all the Figs. 10a, 10b, 10c, and 10d. Please refer to the revised manuscript in detail.

Please also note the supplement to this comment: http://www.atmos-chem-phys-discuss.net/acp-2017-165/acp-2017-165-AC2-supplement.pdf

[Figure]

**Supplement:**

*Supplement of*

**Measurement of scattering and absorption properties of dust aerosol in a Gobi farmland region of northwest China—a potential anthropogenic influence**

Jianrong Bi, Jianping Huang, Jinsen Shi, Zhiyuan Hu, Tian Zhou, Guolong Zhang, Zhongwei Huang, Xin Wang, and Hongchun Jin

Key Laboratory for Semi-Arid Climate Change of the Ministry of Education, College of

Atmospheric Sciences, Lanzhou University, Lanzhou 730000, China

*Correspondence to:* Jianping Huang (hjp@lzu.edu.cn)

[Figure]

38

39

40

41

42

[Figure]

43

**Figure S1.** A variety of agricultural cultivations in Dunhuang farmland (40.492°N, 94.955°E, altitude: 1061 m) prior to the growing season (i.e. from 1 April to 10 May, 2012), producing massive soil dust in the source and downwind regions. (a) The deployment of SACOL's Mobile Facility (SMF) and its adjacent bare farmlands. A tractor was plowing in the nearby farmland on 12 April 2012. (b) Land planning at the afternoon on 20 April, 2012, for the furrow-irrigated land preparation. (c) A ploughing tractor generated a great amount of tiny soil particles into the

atmosphere at the forenoon on 2 May, 2012. (d) An open-cabin tractor was laying plastic mulch nearby the SMF at the afternoon on 6 May, 2012. (e) Land disking for planting at the afternoon on 6 May, 2012. (f) The crops in Dunhuang farmland (nearby SMF) gradually become green on 14 May, 2012, indicating the coming of growth season.

[Figure]

**Figure S2.** Diurnal variations of 10-second average relative humidity (RH, %) under completely clear–sky conditions (14 May, 29 May, and 9 June) and dust events (30 April and 10 June) in Dunhuang farmland. The RH and other meteorological variables were observed by a weather transmitter (Model WXT-520, Vaisala, Finland).